# Anti-Arthritic Activities of Supercritical Carbon Dioxide Extract Derived from Radiation Mutant *Perilla Frutescens* Var. *Crispa* in Collagen Antibody-Induced Arthritis

**DOI:** 10.3390/nu11122959

**Published:** 2019-12-04

**Authors:** Chang Hyun Jin, Yangkang So, Hyo-Young Kim, Sung Nim Han, Jin-Baek Kim

**Affiliations:** 1Advanced Radiation Technology Institute, Korea Atomic Energy Research Institute, Jeongeup-si, Jeollabuk-do 56212, Korea; chjin@kaeri.re.kr; 2Institute of Natural Cosmetic Industry for Namwon, Namwon, Jeonbuk 55801, Korea; yangkang@ncn.re.kr; 3Department of Agricultural Biology, National Institute of Agricultural Science, Rural Development Administration, Wanju 55365, Korea; khy5012@korea.kr; 4Department of Food and Nutrition, College of Human Ecology, Seoul National University, 1 Gwanak-ro, Gwanak-gu, Seoul 08826, Korea; snhan@snu.ac.kr

**Keywords:** collagen antibody-induced arthritis, radiation mutant, *Perilla frutescens*, supercritical carbon dioxide extraction

## Abstract

We investigated the anti-arthritic effects of the radiation mutant *Perilla frutescens* var. *crispa* leaf extract (SFE-M) and wild type leaf extract (SFE-W), both prepared by supercritical carbon dioxide (SC-CO_2_) extraction, on collagen antibody-induced arthritis (CAIA) in Balb/c mice. Animals were randomly divided into four groups: control, CAIA, CAIA + SFE-M (100 mg/kg/day), and CAIA + SFE-W (100 mg/kg/day). The mice were subjected to the respective treatments via oral gavage once daily for 4 days. Mice treated with SFE-M developed less severe arthritis than the CAIA mice. They showed significantly improved arthritic score, paw volume, and paw thickness compared to the CAIA mice from days 3 through 7. Furthermore, histopathological analysis of ankle for inflammation showed that SFE-M treatment reduced inflammatory cell infiltration and edema formation. Similarly, the neutrophil-to-lymphocyte ratio (NLR) in the whole blood was 37% lower in mice treated with SFE-M compared with the CAIA mice. However, treatment with SFE-W did not result in any significant difference compared with the CAIA group. In conclusion, SFE-M treatment delays the onset of arthritis and alleviates its clinical manifestations in CAIA mice.

## 1. Introduction

*Perilla frutescens* (L.) Britt. is an annual herbaceous plant that belongs to family Lamiaceae. Its leaves are used as food in Asian cuisines, and its seeds are used to make edible oil in Korea. Traditionally, *P. frutescens* has also been used to treat a variety of illnesses, including cough, phlegm, back pain, and diabetes [1,2]. In previous studies, extracts derived from *P. frutescens* var. *crispa* were obtained using various methods to analyze various pharmacological activities. For example, both the ethanol extract [3] and the supercritical carbon dioxide (SC-CO_2_) extract showed anti-inflammatory effects. Both water and ethanol extracts exhibited antioxidant effects [4]. The methanol extract exerted a preventive effect against Alzheimer’s disease [5]. In this experiment, we used the extract obtained by SC-CO_2_ method to acquire maximum anti-inflammatory ingredients from the leaves of *P. frutescens* [6]. SC-CO_2_ extraction is a novel and powerful technique for the extraction of lipophilic components [7,8]. Furthermore, SC-CO_2_ extraction is associated with several advantages, compared with the use of organic solvents, because CO_2_ is non-toxic, non-reactive, non-corrosive, and inexpensive. 

Rheumatoid arthritis (RA) is a systemic autoimmune disease, in which chronic joint inflammation leads to cartilage destruction and bone erosion [9]. Typically, RA is treated with pharmacological and non-pharmacological therapies. In the early course of the disease, the pharmacological treatment of RA aims to prevent exacerbation of the disease, using anti-rheumatic drugs [10]. However, in the later stages of the disease, the use of standard drugs in RA induces significant treatment-related side effects. Therefore, the renewed interest in phytoremedies that lack severe side effects and have millennia-proven efficacy is growing [11]. These remedies may have a beneficial effect not only on the symptoms, but also on the pathogenesis of the disease [12]. In this experiment, we sought to determine whether radiation mutant *P. frutescens* could be used as a phytomedicine to alleviate RA.

Mutation induction and selection have been powerful tools in plant breeding, as well as in molecular physiology studies, for the past 80 years. X-ray and γ (gamma) ray irradiation, as well as chemical treatments, have been used for mutation breeding in a wide range of plants [13]. Over the past 40 years, the use of γ ray in mutation induction has increased, while the use of X-ray has significantly decreased. Gamma ray is a type of ionizing radiation that interacts with atoms to induce free radicals in cells, resulting in damage to or modification of important cell and nuclear components of cells, such as chromosomes. The mutant *P. frutescens* var. *crispa* used in this study was also acquired using gamma rays.

In a previous report, the radiation mutant *P. frutescens* var. *crispa* showed enhanced anti-inflammatory activities compared with wild type [14]. Furthermore, the extract from the radiation mutant *P. frutescens* var. *crispa* (SFE-M) obtained by SC-CO_2_ extraction exhibited higher anti-inflammatory activities in RAW264.7 cells compared with the extract derived from wild type (SFE-W) [15]. Although the evidence strongly suggested that SFE-M exerts anti-inflammatory effects, the therapeutic effects of SFE-M on inflammatory diseases such as RA have yet to be investigated. Therefore, the present study was conducted to investigate the effect of SFE-M on RA in an animal model of collagen antibody-induced arthritis (CAIA).

## 2. Materials and Methods 

### 2.1. Animals

Animals were maintained and the study was conducted in accordance with the guidelines of the Guide for the Care and Use of Laboratory Animals (Institute of Laboratory Animal Resources, Korea Atomic Energy Research Institute(KAERI)-IACUC-2017-016). Male Balb/c mice (4 weeks) were purchased from Orient Bio Inc. (Seongnam, Korea) and allowed to acclimate for 1 week prior to the initiation of the study. Mice were maintained in a room under controlled light/dark cycle (12 h/12 h), temperature (about 23 ± 2 °C), and humidity (55 ± 10%).

### 2.2. SC-CO_2_ Extraction

A laboratory-scale supercritical fluid extraction system (Ilshin Autoclave Co., Daejeon, Korea) was used for the SC-CO_2_ extraction of perilla leaves (radiation mutant and wild type). The dried perilla leaves were ground using a milling machine, and the powder (180 g) was transferred to an extraction column. The moisture content in the powder sample was found to be 5.3 ± 1.4%. The powder sample was held in place within the extraction column by glass wool mounted on both ends of the extractor. After the extractor reached the predetermined temperature (50 °C) and pressure (400 bar), the sample was allowed to stand for 10 min for temperature (50 °C) and pressure (400 bar) equilibration. The extraction was performed by passing CO_2_ (99.9%) through the column at a flow rate of 60 mL/min at 50 °C and 400 bar for 3 h. The extracted oil was separated by pressure reduction and collected in the trap. The collected oils were refrigerated at 4 °C. The SC-CO_2_ extraction was repeated twice.

### 2.3. HPLC Analysis

HPLC analysis was conducted using the Agilent Technologies model 1100 instrument (Agilent Technologies, Santa Clara, CA, USA). The samples were analyzed by reverse-phase (C18) HPLC (YMC-Triart C18, 4.6 mm × 250 mm I.D, S-5 μm, flow rate 1 mL/min, UV detection: 254 nm) using acetonitrile:water (44:55 to 55:45, 30 min) as the gradient solvent. Solvents used in HPLC analysis were obtained from Sigma Chemical Co. (St. Louis, MO, USA), and were of analytical grade (≥99.9%).

### 2.4. Sample Preparation and Treatment

SFE-M and SFE-W were suspended in corn oil at a concentration of 20 mg/mL and treated with 100 μL per mouse via oral gavage. Mice were fasted at 7 p.m. and were fed at 10 a.m. after oral administration from days 2 to 6. 

### 2.5. Collagen Antibody-Induced Arthritis

Mice were randomly divided into 4 groups; (1) control (*n* = 6), (2) CAIA (*n* = 6), (3) CAIA plus SFE-M (100 mg/kg, *n* = 6), and (4) CAIA plus SFE-W (100 mg/kg, *n* = 6). A cocktail of four monoclonal antibodies to type II collagen (ArthritoMab; MD Bioscience, Saint Paul, MN, USA; 2 mg/100 μl) was injected intravenously on day 0. Mice in the control group were injected with an equal volume of corn oil. On day 3, all animals except the control group were intraperitoneally injected with LPS (*Escherichia coli* 055:B5; MD Biosciences; 50 μg/200 μl endotoxin-free water). Treatments (corn oil, SFE-M, and SFE-W) were administered by oral gavage once a day from day 3 to day 6. Mice were examined for the development of arthritis for 4 days after LPS injection. 

### 2.6. Assessment of Clinical Signs of Inflammation

Paw volumes were measured using a plethysmometer (Panlab, S.L.U., Digital Water Plethysmometer LE7500, Spain) every day, after LPS injection. The hind leg was soaked in the buffer calibrated with 1 mL standard sinker, and the increased volume was measured. The average volume of both hind legs was used. Paw thickness was measured using a digital caliper (Mitutoyo, Andover, UK) every day, after LPS injection. The average thickness of both hind legs was used. Arthritic score was evaluated blindly using a system based on the number of inflamed joints in the front and hind paws. Inflammation was defined by swelling and redness on a scale from 0 (no redness and swelling) to 3 (severe swelling with joint rigidity or deformity; maximal score for four paws, 12). 

### 2.7. Histopathological Assessement

Hind paws were removed after euthanasia and fixed using 4.5% buffered formalin. Hind paws were decalcified in buffered formalin containing 5.5% EDTA. Upon decalcification, paws were embedded in paraffin (wax) to create paraffin blocks, which were sectioned and stained with hematoxylin and eosin for microscopic evaluation by an expert blinded to the treatments received. Each section was screened for synovial tissue infiltrated by neutrophils and every joint was scored as follows: 0, normal; 1, minimal; 2, mild; 3, moderate; and 4, marked. 

### 2.8. Analysis of Neutrophils and Lymphocytes 

Neutrophil-to-lymphocyte ratio (NLR) is a useful marker for the evaluation of inflammatory activity in chronic inflammatory disease such as ulcerative colitis [16], prostate cancer [17], and RA [18]. Whole blood samples were collected via cardiac puncture. The blood was placed in Vacutainer ^TM^ tubes containing EDTA (BD science, Franklin Lakes, NJ, USA). Anti-coagulated blood was used for hematological testing including neutrophil and lymphocytes in a HEMAVET 950 (Drew Scientific Inc., Miami Lakes, FL, USA). 

### 2.9. Statistical Analysis

One-way analysis of variance (ANOVA) was used to determine the overall differences among groups, followed by Fisher’s LSD test for individual group comparisons. The results from all comparisons were considered significant at *P* < 0.05. Data were reported as mean ± SD. All data were analyzed using the SPSS 21.0 program (SPSS Inc., Chicago, IL, USA).

## 3. Results

### 3.1. Composition of SFE-M and SFE-W

The leaf extracts from radiation mutant *P. frutescens* var. *crispa* and wild type were acquired using the SC-CO_2_ method. Figure 1 shows the composition of the two extracts. The isoegomaketone (IK) content in SFE-M was 76.0 ± 0.7 mg/g, approximately 7-fold higher compared with the IK concentration (10.8 ± 0.3 mg/g) in SFE-W. 

### 3.2. Effects of SFE-W and SFE-M Treatments on the Development of RA in CAIA Model

Initially, whether SFE-M or SFE-W treatment by oral gavage prevented initiation of disease in Balb/c mice with CAIA was investigated. SFE-M-treated mice developed less severe arthritis (Figure 2). Redness and swelling of joints observed in the CAIA group were significantly attenuated with the SFE-M-treated (100 mg/kg). Histopathological examination also revealed that SFE-M treatment reduced synovial hyperplasia and inflammatory cell infiltration in the joint space (Figure 2). The mean histopathological arthritic scores of CAIA-group, SFE-M-treated group, and SFE-W-treated group were (2.33 ± 0.82), (0.00 ± 0.00), and (1.00 ± 0.89), respectively (Table 1 and Figure 3). 

### 3.3. Effects of SFE-W and SFE-M Treatments on Paw Volume in CAIA Model

To evaluate whether SFE-W and SFE-M affected the progression of RA in the CAIA model, male Balb/c mice were gavaged with corn oil with or without SFE-M and SFE-W once a day between days 3 and 6. The CAIA group showed a significant increase in paw volume on days 5, 6, and 7 (17.3%, 14.4% and 20.7%, respectively) compared with the control group (Figure 4). Paw volume was significantly lower in the SFE-M-treated group compared with the control CAIA group on days 5, 6, and 7 (17.4%, 22.8%, and 22.4%, respectively). However, SFE-M treatment did not result in a significant difference in paw volume compared with the CAIA group. 

### 3.4. Effects of SFE-W and SFE-M Treatments on Paw Thickness in CAIA Model

To evaluate whether SFE-W and SFE-M had an effect on RA progression in the CAIA model, the paw thickness was measured by digital caliper. The CAIA group showed significant increase in paw thickness on days 6 and 7 (12.5% and 7.8%, respectively) compared to the control group (Figure 5). Paw thickness was significantly lower in the SFE-M-treated group compared with the control CAIA group on days 5, 6 and 7 (4.7%, 15.3%, and 15.9%, respectively). However, SFE-W-treatment did not result in a significant difference in paw thickness compared with the CAIA group. 

### 3.5. Effects of SFE-W and SFE-M Treatments on Arthritic Score in CAIA Model

Arthritic score was determined blindly by three persons to further evaluate whether SFE-M and SFE-W treatments suppressed RA progression in the CAIA model. The CAIA group showed a significant increase in arthritic score from days 4 through 7 compared with the control group (Figure 6). The CAIA group showed arthritic symptoms in all joints from day 4 through day 7. The arthritic symptoms were significantly attenuated in the SFE-M-treated group on days 4 through 7. However, SFE-W-treatment did not result in a significant difference in arthritic score compared with the CAIA group.

### 3.6. Effects of SFE-W and SFE-M Treatments on Blood Cell Population in CAIA Model

NLR is a measure of the absolute neutrophil count relative to lymphocyte numbers in the whole blood. To further determine whether SFE-M and SFE-W treatments affect blood cell population in the CAIA model, NLR was measured in the whole blood sample. The CAIA group showed a significant increase in NLR on day 7 compared with the control group (Figure 7). The NLR level was lower in the SFE-M-treated group compared with the CAIA group by 37%. However, SFE-W-treatment did not result in a significant difference in NLR levels compared with the CAIA group.

## 4. Discussion

In the present study, the anti-arthritic effects of the extract from radiation mutant *P. frutescens* var. *crispa* prepared via supercritical carbon dioxide extraction (SFE-M) on the development of arthritis in CAIA model was investigated. The efficacy of SFE-M was compared with that of SFE-W. Treatment with SFE-M alleviated immune cell infiltration into joint synovium, paw edema, arthritic score, and NLR levels. Treatment with SFE-W had no significant effect on the development of arthritis. In a previous study, SFE-M showed higher anti-inflammatory activities than SFE-W in LPS-stimulated RAW264.7 cells [15]. Higher anti-inflammatory activities of SFE-M appeared to be due to the higher IK content (almost 7-fold higher level) compared with SFE-W [15]. Similar to the previous study, SFE-M was more effective in delaying the onset of arthritis in CAIA model, compared with SFE-W. 

Radiation-induced mutations have been extensively studied and utilized in mutation breeding, following the discovery of X ray-induced mutations in *Drosophila* [19] and barley [20]. Later, it was found that ionizing radiation induced DNA damage, which is a major factor underlying mutations [21]. Radiation-induced mutation breeding focused on crop improvement [22], enhancing resistance to abiotic and biotic stresses [23], and development of new flower varieties [24]. However, increased functional metabolites in radiation-induced plant mutants has never been reported. This study is significant, in that it provides the evidence for the possibility of therapeutic use of radiation-induced plant mutants by increasing functional phytochemical content. One of the biggest challenges in the investigation of functional foods or phytomedicines using natural resources is that natural resources usually contain very small amounts of functional components. In this study, the possible selection of new resources containing higher levels of functional constituents derived from radiation-induced plant mutants was confirmed. The radiation-induced mutant *P. frutescens* var. *crispa* used in this study was acquired using gamma rays. It contained about 7–fold higher IK than the wild type species. IK is biosynthesized from egomaketone (EK), and this reaction is inhibited by gene I in *P. frutescens* [25]. Therefore, it seems that gene I was affected by gamma radiation, and consequently exhibited lower activity, compared with wild type. The correlation between gene variation and changing IK content is currently being studied. The amount of IK in 100 mg SFE-M is about 6.38 mg. However, treatment with SFE-M (100 mg/kg) showed more effective anti-arthritic action compared with pure IK treatment (10 mg/kg), in CAIA animal model [26]. This may be attributed to other ingredients contained in the extracts, in addition to IK.

Rheumatoid arthritis (RA) is a systemic autoimmune disease in which chronic joint inflammation leads to cartilage destruction and bone erosion [9]. Generally, the use of standard drugs in RA triggers numerous side effects such as infusion hypersensitivity reactions, nausea, dry mouth, somnolence, fatigue, and severe infection [27,28,29]. Currently, there is a growing interest in medicines of botanical origin, which lack severe side effects and have proven efficacy in traditional medicine [11]. These remedies may be effective not only in alleviating the symptoms but also in ameliorating the pathogenesis of the disease [12]. Several anti-arthritic medicinal plants have been tested in animal and human studies: *Arnica montana* [30], *Boswellia* spp. [11]; *Curcuma* spp. [31]; *Equisetum arvense* [32]; *Harpagophytum procumbens* [33]; *Salix* spp.; and *Sesamum indicum* [34]. Radiation-induced mutant *P. frutescens* var. *crispa* used in this study exhibited higher anti-inflammatory activities, compared with wild type. Its extract obtained by supercritical carbon dioxide extraction also displayed adequate potential as an anti-arthritic medicinal plant. To the best of our knowledge, this is the first report that describes radiation-induced plant mutants containing higher anti-arthritic properties, compared with wild type species. However, this experiment was conducted only with male mice; but given that women are more likely to have arthritis than man [35], it will be necessary to use female mice, before applying the treatments to humans. Moreover, it was reported that female mice carrying the *Cia40* congenic locus are more affected by collagen-induced arthritis, than are male mice [36].

## 5. Conclusions

The extracts of radiation mutant *P. frutescens* var. *crispa* (SFE-M) obtained via SC-CO_2_ method contained 7-fold higher IK content than the extracts obtained from wild type (SFE-W). SFE-M showed anti-arthritic activity in CAIA model, unlike SFE-W. Therefore, it is thought that radiation mutant *P. frutescens* var. *crispa* may be a potential candidate for the treatment of inflammatory diseases, such as RA.

## Figures and Tables

**Figure 1 nutrients-11-02959-f001:**
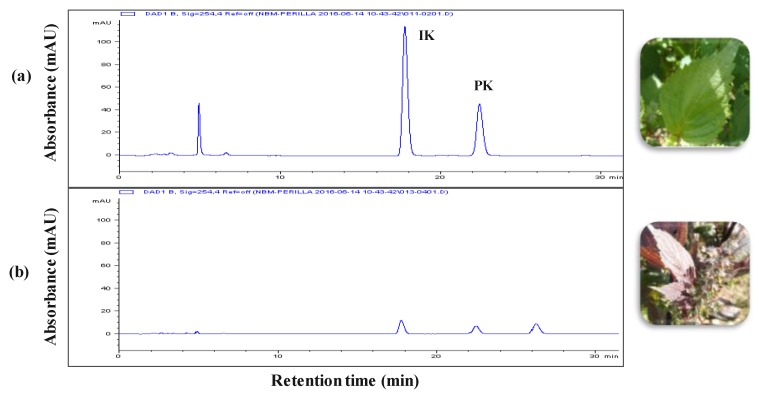
HPLC chromatograms. (**a**) radiation mutant *Perilla frutescens* var. *crispa* leaf extract prepared by supercritical carbon dioxide extraction (SFE-M) and (**b**) wild type leaf extract prepared by supercritical carbon dioxide extraction (SFE-W); IK: isoegomaketone, PK: perillaketone.

**Figure 2 nutrients-11-02959-f002:**
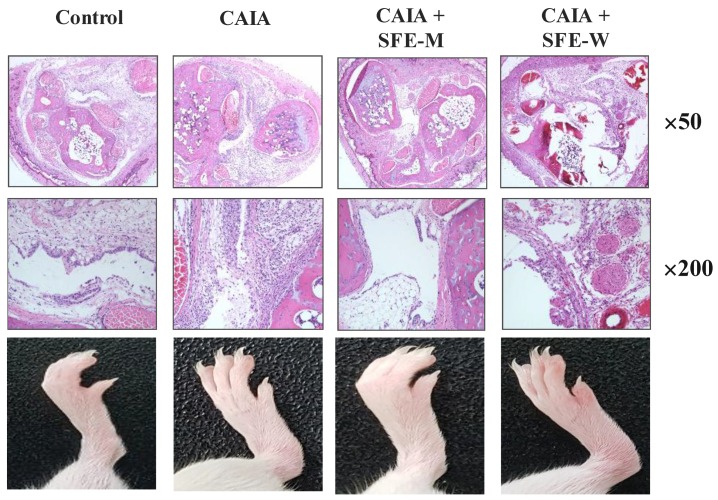
Representative microscopic images of knees and joints. Treatment concentrations of SFE-M and SFE-W were 100 mg/kg. Corn oil, SFE-M, and SFE-W were administered via oral gavage once per day from day 3 to day 6.

**Figure 3 nutrients-11-02959-f003:**
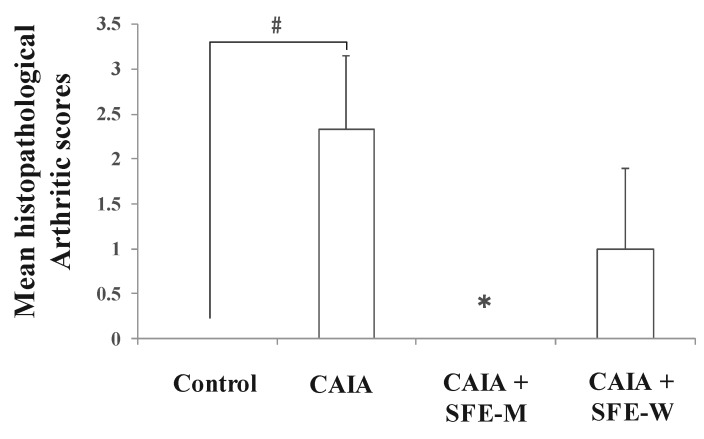
Effects of SFE-M and SFE-W treatments on the mean histopathological arthritis score in collagen antibody-induced arthritis (CAIA) mice. Treatment concentrations of SFE-M and SFE-W were 100 mg/kg. Results were expressed as a score (mean ± SD) of six mice. ^#^
*p *< 0.05 vs. control group, and * *p *< 0.05 vs. CAIA group. Joints were scored as follows: 0, normal; 1, minimal; 2, mild; 3, moderate; and 4, marked.

**Figure 4 nutrients-11-02959-f004:**
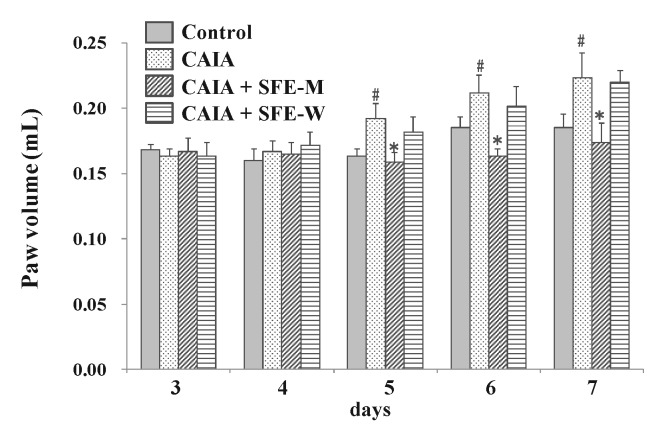
Effects of SFE-M and SFE-W treatments on paw volume in CAIA mice. Treatment concentrations of SFE-M and SFE-W were 100 mg/kg. Paw volume was measured by digital plethysmometer every day after LPS injection and oral treatments. The average volume of both hind legs was used. Data are presented as mean ± SD (*n* = 6). ^#^
*p *< 0.05 vs. control group and * *p *< 0.05 vs. CAIA group.

**Figure 5 nutrients-11-02959-f005:**
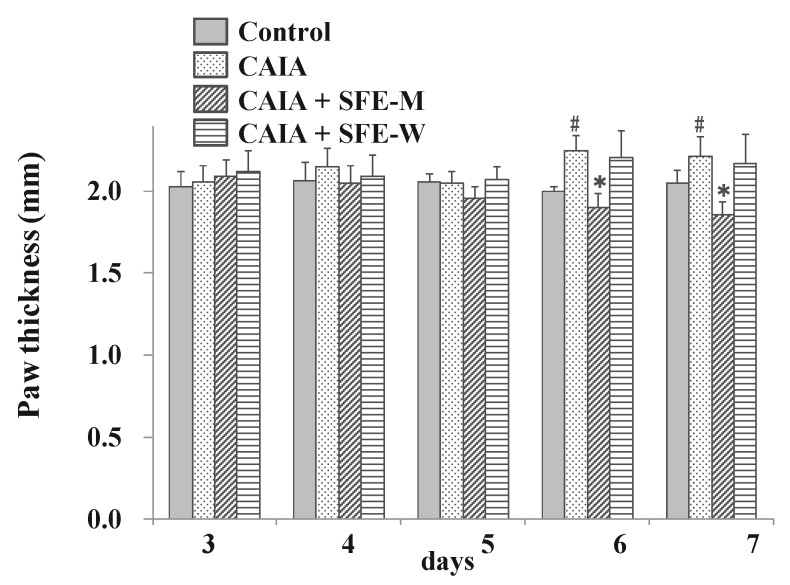
Effects of SFE-M and SFE-W treatments on paw thickness in CAIA mice. Treatment concentrations of SFE-M and SFE-W were 100 mg/kg. Paw thickness was measured using a digital caliper every day, after LPS injection and oral administration of treatments. The average thickness of both hind legs was used. Data are presented as mean ± SD (*n* = 6). ^#^
*p *< 0.05 vs. control group, and * *p *< 0.05 vs. CAIA group.

**Figure 6 nutrients-11-02959-f006:**
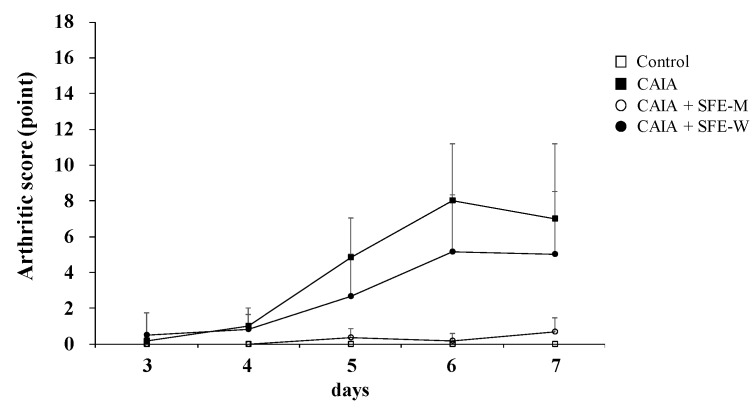
Effects of SFE-M and SFE-W treatments on arthritic score in the CAIA mice. Treatment concentrations of SFE-M and SFE-W were 100 mg/kg. Arthritic score was evaluated blindly using a system based on the number of inflamed joints in the front and the hind paws. Data are presented as mean ± SD (*n* = 6). SFE-M-treated mice show significantly lower severity than CAIA mice (*p* < 0.05).

**Figure 7 nutrients-11-02959-f007:**
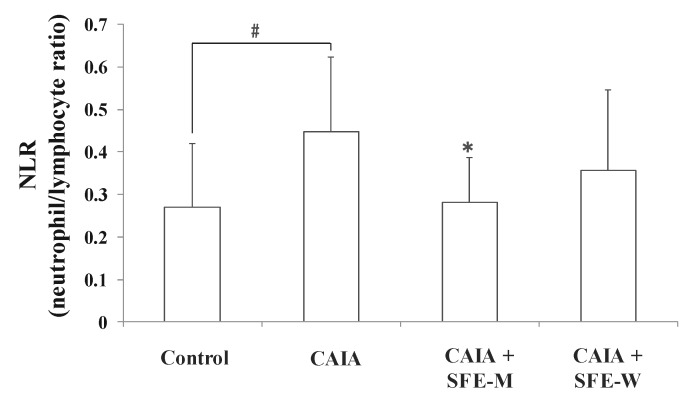
Effects of SFE-M and SFE-W treatments on neutrophil-to-lymphocyte ratio in CAIA mice. Treatment concentrations of SFE-M and SFE-W were 100 mg/kg. Whole blood samples were collected by cardiac puncture. Data are presented as mean ± SD (*n* = 6). ^#^
*p *< 0.05 vs. control group and * *p *< 0.05 vs. CAIA group.

**Table 1 nutrients-11-02959-t001:** Histopathological scores of the groups.

Organ	Group		Control	CAIA	CAIA + SFE-M	CAIA + SFE-W
Ankle joint	Inflammation	-	6	0	6	2
		±	0	1	0	2
		+	0	2	0	2
		++	0	3	0	0
		+++	0	0	0	0

Grade: -: normal, ±: minimal, +: mild, ++: moderate, +++: marked. No. of examined: 6/group.

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
