# Peer review of "Anti-Arthritic Activities of Supercritical Carbon Dioxide Extract Derived from Radiation Mutant Perilla Frutescens Var. Crispa in Collagen Antibody-Induced Arthritis"

_nutrients, 2019, doi:10.3390/nu11122959_

Round 1

Reviewer 1 Report

The authors have made a case for radiation mutant of P. frutescens var. crispa as a potential anti-arthritic and in general anti-inflammatory drug candidate. The introduction is adequate and the problem has been clearly stated. The authors have provided sufficient data that corroborates their claim. Though I overall agree with authors' conclusions, I would recommend to rephrase the last statement in conclusion in which authors have claimed that the extracts from the mutant plant can be used to treat anti-inflammatory diseases. The authors have provided evidence for only anti-arthritic properties of this mutant and do not present any additional data regarding any other inflammatory diseases and therefore should refrain from making generic statements. 

Author Response

Dear reviewer:

We are grateful for your thoughtful correction for manuscript (Nutrients-647740).

Sincerely yours.

Jin-Baek Kim.

Reviewer 2 Report

The introduction and discussion were not written well, need more explanation with current references. In method section how the author induced CAIA was not well described. Also the necessity of LPS injection in the method section is also missing. Body weight of the mice was not mentioned clearly in the method section. Why female mice were excluded from the experiment? Author should use equal gender and number of mice for this experiment. Data representation is poor.  In figure 6 and 7, The error bar are so big, but still statistically significant. Please justify.

Author Response

(The authors gave the same response as above.)

Reviewer 3 Report

As far as I am concerned the manuscript is well written. The subject area of research is significant in knowledge development. The introduction is interesting and correct. The discussion is well conducted supported by other results. The literature is well-chosen, although there could also be newer items. Graphic representation of results correct,although the caption for figure 5 is on the next page.

Author Response

(The authors gave the same response as above.)

Round 2

Reviewer 2 Report

The authors didn't convenience or corrected properly any of the queries except comment2. eg,

in comment1: "The introduction and discussion were not written well, need more explanation with current reference.

Answer: Thank you so much for your valuable and important comment. But, I couldn’t revise because I didn’t know what you pointed out. Can you tell me more detail?"

My suggestion was clear that the writing style for introduction and discussion didn't put much interest, need strong way of writing. How they will do it, that is not possible for a reviewer to explain. 

In comment 3: "female mice were excluded from the experiment? Author should use equal gender and number of mice for this experiment.

Answer: Thank you so much for your valuable and important comment. To our knowledge, female mice are generally not used in experiments because of the potential for hormones to affect experimental results."

This is an old concept, now a day both sex are being used for experiment, not anymore. And if you further think about the clinical implication, then female are suffering more in arthritis and they should be considered as well in the study. If the whole experiment in mice exclude female, then in near future if the author will go 1-2 step ahead for human sample or human trail, then what will be the scenario?

I am giving one example in arthritis using female, but there will be more. Not only arthritis, previously every studies exclude female rat or mice, but not anymore (pubmed serach will be helpful for sure).

Arthritis Res Ther. 2008; 10(4): R88.

Published online 2008 Aug 6. doi: 10.1186/ar2470

PMCID: PMC2575638

PMID: 18684326

Increased susceptibility to collagen-induced arthritis in female mice carrying congenic Cia40/Pregq2 fragments

Maria Liljander,1 Åsa Andersson,2 Rikard Holmdahl,3,4 and Ragnar Mattsson1

Author information Article notes Copyright and License information Disclaimer

In comment 4: "Data representation is poor. In figure 6 and 7, the error bar are so big, but still statistically significant. Please justify.

Answer: Thank you so much for your valuable and important comment. I agree that the margin of error is large. But, statistical values were obtained by ANOVA analysis."

Here I wanted to know how they got the significance with such a high error, they should explain it properly in the author’s response. And they also missed the first line: “Data representation is poor.” My suggestion is that author should work on their data representation, they can get an idea from other publications in this journal.

In comment 5: "English language and style: Extensive editing of English language and style required.

Answer: We edited our manuscript through professional English editors."

That doesn’t justify the English correction suggestion by the reviewer. Even if it was editing by professional person, but if it still not meet the level of the journal’s writing quality and style, the author should work more onto it.

Author Response

Dear Reviewer.

Thank you for your valuable comments.

According to your comments, we tried to revise manuscript.

We hope that the revised part will meet your expectations.

Sincerely yours.

Jin-Baek Kim
